# Sustainable Electromagnetic Prototype for Detecting Internal Deterioration in Building Walls

**DOI:** 10.3390/s24144705

**Published:** 2024-07-20

**Authors:** Papa Pio Ascona García, Marco Antonio Aguirre Camacho, Elger Orlando Silva Barboza, Guido Elar Ordoñez Carpio, Emerson Cuadros-Rojas

**Affiliations:** 1Faculty of Engineering, Universidad Nacional Intercultural Fabiola Salazar Leguía de Bagua, Jirón Ancash 520, Bagua 01721, Amazonas, Peru; maguirre@unibagua.edu.pe (M.A.A.C.); esilva@unibagua.edu.pe (E.O.S.B.); 2Facultad de Ingeniería Civil y Ambiental, Universidad Nacional Toribio Rodríguez de Mendoza de Amazonas, Calle Higos Urco N° 350, Chachapoyas 010001, Amazonas, Peru; guido.ordonez@untrm.edu.pe; 3Department of Civil and Environmental Engineering, Universitat Politècnica de Catalunya (UPC-BarcelonaTech), Jordi Girona 1-3, 08034 Barcelona, Spain; emerson.julio.cuadros@upc.edu

**Keywords:** prototype of electromagnetic detector, interior deterioration of structural cladding, densities of building walls

## Abstract

The aim of this study was to develop a sustainable electromagnetic prototype to detect the interior deterioration of walls in buildings in order to mitigate uncertainty as it is a challenge to observe the interior state of walls without utilising destructive procedures. The method used was experimental, developmental and quantitative in its approach. The inductance, electric current, modulated frequency and power of the electromagnetic field were used to penetrate the constructed specimens, which were built of materials such as concrete, brick, adobe, plaster and fine sand and had walls with a thickness of less than 300 millimetres. The results show that the optimum value of the magnetic field was 0.18 µT, which was sufficient to penetrate 150 mm with densities between 1.0 and 2.4 g/cm^3^ and porosities between 11 and 60%. The current and wave each had a coefficient of determination R^2^ = 0.8914, and the average inductance value was 184 µH, which was established with an air core of radius 9.75 cm and with 19 turns with AWG-25 wire. The frequency-modulated signal ranged in the audible zone between 10 and 22 kHz. The presented prototype detects the interior deterioration of the walls of the building, and the signal is reflected on a metallic guide on the opposite side of the wall with a reading error of 5%. The use of this prototype does not represent a risk to the operator or the environment.

## 1. Introduction

The problem presented in this study is aggravated in the case of self-constructed buildings, which are increasing in poor regions due to a shortage of resources, lack of control, real estate pressure and high housing prices. These self-constructed buildings are exposed to physical, chemical and mechanical failures in their walls and ceilings. This context generates uncertainty for the occupants of these houses regarding the safety of the buildings, even more so in seismic zones in which such buildings are prone to collapse, incurring material and economic losses. Therefore, the early detection of deterioration and the constant monitoring of a building’s structural walls can help in making decisions to maintain or increase the life span of a building. 

Internal cracks and fissures, the main cause of structural damage, are usually below the surface of building facades and are difficult to detect by surface observation, which jeopardises early intervention and increases the vulnerability of the structure. This phenomenon is especially common in older buildings [1]. The occurrence of internal damage can be attributed to various factors, such as the deterioration of materials over time, structural stress, moisture, poor quality of building materials, improvised workmanship of builders, reactions of salts contained in the soil with the surfaces, thermal shrinkage of concrete structures and overloading of concrete surfaces [2]. These factors contribute to a reduction in the service life of a building. The concept of service life is an essential component of modern structural engineering and is determined by factors such as design, construction, aging and maintenance during the service life of a structure [3]. The guidelines for determining the service life of structures are described in the design and construction standards [4] and supplemented by [5].

The detection of internal damage is undoubtedly the most relevant parameter in the assessment of damage in structural engineering. The advance of technological research has had a disruptive effect in recent years and has made possible the manufacture of various sensors and a concomitant expansion of methods, among which non-destructive testing (NDT) methods are gaining prominence [6]. These methods allow for the evaluation of a structure while it is still operational, and this is defined as a process that inspects the quality and properties of materials and products without altering the functioning of the building [3]. The main non-destructive testing methods are classified into categories such as acoustic methods, radar methods, electromagnetic methods, penetrating radiation, physical methods, thermal methods and other methods (X-ray tomography, etc.). However, the detection of internal defects in walls can be affected by different types of non-invasive methods, which highlights the need for user uncertainty analysis.

Among these methods, magnetic sensor methods can be used as an alternative method and are currently used in many applications, with the magnetic field that is used providing values in micro-Tesla (µT) units [7]. Also, magnetic multilayers are able to provide important sensitivity, are used as magnetic field detectors and can be used for density measurement [8]. However, in most of these instances, the range is limited to 25 mm, which is inadequate for structural walls that typically exceed a 150 mm thickness when constructed with concrete walls and can be larger when other types of materials are used. 

Therefore, the main objective of our study was to develop a sustainable electromagnetic prototype to detect the interior deterioration of the cladding in buildings, to demonstrate the extent of the penetration of the electromagnetic field in the density and porosity of different types of common materials used in the construction of buildings and to indicate the value of the inductance and number of turns required to generate the optimum magnetic field, increasing the current in transit so that the induction of the electromagnetic wave is at a maximum and analysing the modulated frequency of the two signals and the electromagnetic spectrum of the prototype.

This prototype design does not affect the structural integrity of the walls it analyses, i.e., it is free from destruction. The assembly process of the equipment was developed using materials and 3D software, and the tests were carried out in the laboratory of the Faculty of Civil Engineering of the Fabiola Salazar Leguía University of Bagua, which allowed for the electromagnetic field prototype (B) to be properly assembled and for tests to be carried out on the six specimens built from different types of materials.

Finally, the constructed prototype demonstrated that it emits a strong electromagnetic field and is detectable at a distance of 300 mm, which is sufficient to penetrate all of the inner layers of walls that are commonly found in residential buildings constructed with adobe, brick, concrete and stone. The readings obtained by the prototype allow for the physical state of the interior of a building’s structural walls to be known in real time and at a low cost. The following two sections show the theoretical framework that supports the operation of the proposed prototype. Also, the authors of [9] note that “scintillation detectors buried underground are used for muon density measurement”, which is consistent with our study.

### 1.1. Generation of Electromagnetic Fields 

A magnetic field can be generated from a varying electric field [10], Ampere’s law represents a specific instance of the Ampere and Maxwell equation, applicable when an electric field is independent of time. Furthermore, the Maxwellian equations enable the prediction of the time evolution of a magnetic field [11]. In [12], the field strength was quantified as a function of its position under the plasmon condition, in which the field strength decays exponentially as one moves away from the metal–vacuum interface [13]. Simulation tools can be used to obtain the return loss, radiation pattern, field strength and realised gain [10]. The magnetic field is generated in the centre of the air core, as shown in Figure 1.
(1)$$B=\frac{N\mu_{0}I}{2\cdot r}$$
where I represents the intensity expressed in amperes (A), while $\mu_{0}$ denotes the permeability of the medium at the centre of the coil (Tm/A). B represents the magnetic induction at the centre of the coil, measured in Teslas (T), r denotes the radius of the coil, measured in metres (m) and N is the number of turns or number of turns on a circumference. 

However, in [14] it was shown that, for a solenoidal receiver coil, the method gives the same results as the traditional calculation method, but its advantage lies in its ability to predict the relationship for other coil configurations, from which Equation (2) can be derived to calculate the magnetic field of a coil.
(2)$$B=\frac{N\mu_{0}I}{2\cdot \sqrt{({\frac{H_{b}}{2})}^{2}+({\frac{D_{b}}{2})}^{2}}}$$
where $H_{b}\;y$ $D_{b}$ are the dimensions of the solenoid, consisting of the radius and the length, which are determined by the number of turns of the wire.

Furthermore, reference is made in [15] to the design and fabrication of a device for measuring the electromagnetic field (EMF) induced by a magnetic field generated by a circular loop. Therefore, the calculation of the EFM using Equation (2) was an important parameter in the design of the prototype. 

The electromagnetic field was designed, simulated and implemented in a coil external to the empty core with the objective of ensuring that the magnetic field (B) is sufficiently strong enough to penetrate the hardness of the object in question. In this case, structural walls was constructed using a variety of materials, with typical thicknesses measured in centimetres. The inductance (L) is an indispensable component of the prototype. The unit of the wire cross-section (H = Henries) is considered for the winding, as is the magnetic permeability (µ) of the material, which is its ability to absorb attractive forces in time and space. These parameters are detailed in the following formulas:(3)$$µ={µ}_{0}\cdot {µ}_{r}$$
(4)$${{µ}_{0}=4\cdot \pi \cdot 10}^{-7}\frac{H}{m}$$
where ${µ}_{0}$ denotes vacuum permeability, and ${µ}_{r}$ is the relative permeability of the material (in this investigation, it is air, i.e., ${µ}_{r}$ = 1).

In order to calculate the inductance of this prototype, the following equation was considered:(5)$$L=4\cdot \pi \cdot 10^{-7}\frac{H}{m}\cdot \left\lfloor \frac{N^{2}\cdot D}{\frac{N\cdot d}{D}+0.44}\right\rfloor$$
where L represents the inductance in Henries (H), while N denotes the number of turns of a 25-gauge AWG enamelled wire. D represents the circumference diameter (mm), d denotes the wire diameter (mm) and π is 3.141592.

In addition, the resonant frequency of each stage of an electromagnetic field generator is given by:(6)$$f_{0}=\frac{1}{2\pi \sqrt{L\cdot C}}$$
where $f_{0}$ denotes the resonant frequency in Hertz, and C represents the capacitance in farads.

The electromagnetic field is regarded as a non-inertial system, derived through a geometrical calculation in which the propagation vector of the electromagnetic wave undergoes a change in direction at each instant of time due to the angular velocity [16].

### 1.2. Structural Deterioration

There is growing interest among building users in enhancing the durability of buildings while simultaneously pursuing a more sustainable lifestyle [17]. It is therefore evident that Structural Health Monitoring (SHM) is a valuable tool for assessing the condition of the built environment and its behaviour in operation. This is due to the fact that the mechanical properties of materials such as masonry are not linear and the fact that their mechanical properties are uncertain due to environmental factors and their internal texture [18].

There are various techniques for measuring the toughness of materials commonly used in building construction. One such method is the SONREB approach, which relates the compressive strength of concrete cylinders to the rebound using a digital sclerometer [19] and DIC [20]. This methodology employs a statistical regression approach to assess the strength of concrete, albeit with a degree of sample destruction. An alternative approach is the ultrasonic pulse velocity method, which also predicts the strength of concrete based on the velocity and a quality index of construction materials. Furthermore, in [21,22], an experimental analysis was conducted for the characterisation of adobe brick masonry as a structural material. In [23], it was found that concretes produced with rubber and metakaolin exhibited bulk density values below 2000 kg/m^3^. Consequently, the objective was to reduce the bulk density and obtain values for the compressive strength and water absorption. The principal techniques employed for the assessment of infrastructure are outlined in Table 1.

The assessment must be accompanied by a study of the causes and effects of the pathologies, and an adapted and summarised methodology is shown in Figure 2. In the case of mechanical failure, cracks play a relevant role in the deterioration of the structural element, and the real cracks are characterised by their spatial curvature [25].

These can be caused by exceeding the mechanical properties due to tensile, compressive, bending and horizontal tension actions [26]. Therefore, some standards limit wall thicknesses; for example, for masonry walls, in the E-070 standard, the effective thickness is represented as t ≥ h/20 for seismic zones 2 and 3, while it is t ≥ h/25 for seismic zone 1, and it is defined by Equation (7).
(7)$$\sigma_{m}=\frac{P_{m}}{L\cdot t}\leq 0.2f_{m}^{\prime }\left[1-{\left(\frac{h}{35\cdot t}\right)}^{2}\right]\leq 0.15\cdot f_{m}^{\prime }$$
where $\sigma_{m}$ represents the maximum axial stress, L denotes the total length of the wall, t represents the effective thickness and $P_{m}$ represents the maximum gravity load. Additionally, $f_{m}^{\prime }$ denotes the quality of the masonry, and h represents the effective height of the element; therefore, an innovative means of detection is required.

Regarding the cracking model in [27], it is mentioned that when longitudinal cracks appear in the cladding of a structure, the safety of the structure is seriously threatened; furthermore, it is essential to assess the adhesion of the cladding and the strength of the concrete walls to provide a more precise classification [28]. Studies on other types of materials, such as adobe, are presented in [29]. A sample of structural failures in dwellings with different types of materials is shown in Figure 3.

**Figure 2 sensors-24-04705-f002:**
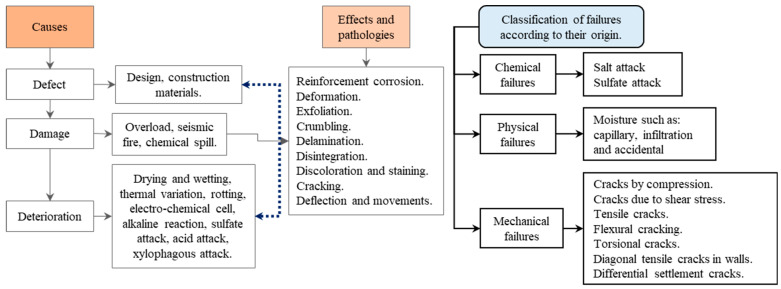
Pathology of structural failure causes and effects. Note: adapted from [30] and supplemented by [31].

In addition, discontinuities in the density of structures may be indicative of a crack or fissure. These are a warning of an event that may compromise the durability and serviceability of the building structure, but they must be taken with caution in buildings constructed using current methods, such as 3D-printed concrete [32].

Conversely, the Rockwell and Brinell hardness test, based on the ASTM standard, is a valuable and effective method for determining the hardness of materials. This is because it measures the depth of penetration of a penetrator (a hardened steel ball of different diameters) into the test specimen (metals and plastics). This enables the relative strength and durability of the materials to be calculated using the Equation (8), as follows.
(8)$$FC=N-\frac{d}{c}$$
where FC represents the Rockwell hardness test, with N denoting a value of from 100 to 130 on the Rockwell scale, d denotes the depth of penetration and c represents a range of from 0.001 to 0.002 mm on the aforementioned scale.

With regard to the structures of residential buildings, the most common failures include inadequate strength and high stresses in walls, columns and beams, as well as shear bond problems, vibration, torsion and punching [33]. Furthermore, a diagnostic tool has been developed that assesses materials, pathologies and seismic vulnerability, which is useful for the retrofitting of old buildings [34]. The incorporation of ARM enhances the physicomechanical attributes of mortars, with the exception of their density, water absorption and porosity. Moreover, the mechanical strengths of mortars exhibit an increase over time [35]. A novel method yields concrete mixtures that are comparable or superior to conventional concrete, while simultaneously reducing cement consumption [36]. A comparison of the mechanical properties of lightweight concretes demonstrated a minimum compressive strength of 17 MPa at 28 days and a maximum density of 1840 kg/m^3^, while the structural concretes exhibited enhanced strength characteristics when utilising alternative aggregates, such as expanded clay, reaching a compressive strength of 32 MPa and a dry density of 1823 kg/m^2^ [37]. Finally, alternatives such as the utilisation of recycled plastic have exhibited enhanced potential in terms of their compressive strength, underscoring their applicability in water retention structures and walkways [38]. This reinforces the necessity for sustainable and efficient technologies in construction.

Therefore, it is necessary to incorporate government investment in energy-efficient technologies as a means of addressing the shortcomings of inadequate housing [39]. This investment would serve to identify a crucial modification for structural earthquake stability, utilising hazard-dependent values for the behavioural factor [40]. It has been observed that social programmes [41] incorporate a sustainable morphology in the urban configuration, which has a positive impact on quality of life. However, challenges persist, including unequal access to basic services and the promotion of economic dependency, which affect social development and the distribution of inhabitants in residential areas. 

## 2. Materials and Methods

In order to design and construct the electromagnetic field equipment, a variety of materials and equipment were employed. These included PVC blocks and tubes and laboratory equipment such as an ultra-spectrum analyser, a digital oscilloscope, a frequency meter, an LCR meter, a computer and a voltmeter.

Additionally, simulator programs such as N-MULTISIM and Proteus were utilised, and supplies such as electronic component sets, printed boards, coils and enamelled copper wires, a power supply and batteries were also used. Instrumental sets were also required. The aforementioned materials and equipment were required for the construction of the internal and external components of the prototype. In order to construct the six test specimens, a set of construction tools were employed, including adobe, concrete, brick and plaster materials. The aforementioned specimens were employed in the prototype tests.

The prototype design comprised the following elements:(a)The design of architectural plans.(b)The determination of the circuit elements and parameters.(c)The simulation of the electronic circuit with the Proteus and MULTISIM software (https://www.ni.com/zh-cn/support/downloads/software-products/download.multisim.html#452133, accessed on 20 June 2024).(d)The installation and connection of electronic devices on the test and printed circuit board.(e)Testing the operational functionality of the prototype with laboratory equipment.

The process of implementing the prototype included the following steps:(a)Structural design and prototype parts.(b)The assembly and construction of the prototype.(c)The construction of specimens utilising adobe, concrete, brick, plaster and fine sand.(d)Conducting operational testing at the site in question.(e)Readjustments and quality control.

The methodology and design of the study aligned with the tenets set forth by [42] and were regarded as experimental in nature due to the deliberate manipulation of the independent variable (VI) to assess structural failure (DV) [43]. This type of innovation in research and development (R&D&I) is applied with the objective of transforming a need into a solution alternative through the utilisation of scientific knowledge, technological means and methodological protocols. In light of the aforementioned AI [44], the conclusions were subjected to hypothetical–deductive analysis, considering the statements put forth by [45]. This study employed a quantitative in approach, with the population and sample being equal, comprising six specimens constructed with different types of materials. The data were collected through the use of observation guides in the test field. A comparison of the density and penetration distance of the magnetic field (B).The circuits, specimens and final product were designed and drawing. Statistical data analysis was performed on the cabinet.

### Contribution

The deterioration of structures, which manifests as cracks and fractures in beams, columns, walls, roofs and floors, can be attributed to a range of factors. These include physical, chemical and mechanical processes as well as water infiltration and the subsequent destruction of other services. The consequence of the failure to address deficiencies in design, improvised workmanship and the use of inadequate materials in construction is an increase in the deterioration of the internal wall, which, in turn, gives rise to greater uncertainty among building users with regard to the safety of the structures in which they work. 

Figure 4 delineates the methodology employed in the fabrication of the sustainable electromagnetic prototype, which is tasked with detecting the interior deterioration of coatings in buildings. This is achieved by modulating the amplitude and the two frequencies in a uniform manner over the course of a varied cycle with a certain degree of deviation, thereby enhancing the penetration range of the magnetic field “B”. In addition to the correlations of the parameters such as the inductance, number of turns and measuring equipment (RCL), the computer program demonstrates a positive correlation when the current (i) in the collector of transit Q1 is increased. This results in the coil L1 generating an increased magnetic field, designated as “B”.

In order to achieve the stated objectives, it was necessary to develop specific activities, as illustrated in Figure 4. This approach allowed for a continuous feedback loop, enabling the team to revisit any stage of the assembly process if necessary to ensure the desired outcomes were met. 

Furthermore, if the measured material properties of the tested specimen were found to be inferior to those indicated in Table 2, this could be indicative of a deterioration in the internal structure of the structure in question. In other words, if the density of the hollow brick was less than 1.1 g/cm^3^, if the density of the fired brick was less than 1.3 g/cm^3^ or if the density of the concrete/cement was less than 2.2 g/cm^3^, the materials were deemed to be deteriorated.

## 3. Results

The design employs mathematical formulas and the MULTISIM program to enhance the confidence in the simulation of the electrical circuit. This enables the determination of key parameters such as the magnetic field generation, oscillation frequency, voltage, current consumption and waveforms at each stage. Table 3 illustrates the calculated inductance (Lc), the measured inductance (Lm), the software-calculated inductance (Ls) and the number of turns in the coil (Nv), see Figure 5. Additionally, Table 4 shows the frequency results measured for each of the six samples.

Furthermore, Figure 6 illustrates the relationship between the magnetic field and the number of spirals, demonstrating that an increase in the number of turns resulted in an elevated electromagnetic wave output. However, this growth was not unlimited (R² = 0.9911, r = 0.9955). Figure 7 illustrates the transistor oscillation frequency, which manifested as a green signal at 47.9 kHz and a blue signal at 54.7 kHz. These signals exhibited a transient coincidence, followed by a 180-degree phase difference. Figure 8 illustrates the amplitude and period of the waves observed on oscilloscope channels A and B, with the emitter of transistors Q1 and Q2 shown in blue and green, respectively. Additionally, Figure 9a presents the modulated waveform at the base of transistor Q3, while (b) shows the initial modulated amplitude increment with a frequency between 2 and 22 kHz. Moreover, Figure 10 presents an electromagnetic spectrum analysis of the prototype, indicating an operating spectrum of 56 kHz with a gain of 10 dB. Figure 11 illustrates the carrier signal with the audio frequency at the Q3 collector, which operates at approximately 10–22 kHz. Figure 12 provides an illustration of the internal components of the electronic circuitry of the prototype. Subsequently, Figure 13 presents the specimen design, with a plan view on the left and a perspective view on the right. Figure 14 illustrates the specimens constructed with varying construction materials, which were tested in open air and comprised adobe, brick, plaster, concrete and fine sand, and they exhibited no discernible interference. Figure 15 depicts the prototype interior wall deterioration detector and its components, which was tested on concrete, brick-with-holes, adobe, gypsum and fine-sand specimens. Furthermore, Figure 16 presents an analysis of the electromagnetic spectrum of the prototype, demonstrating that it was within the established non-ionising frequency range. Figure 17a provides a detailed account of the current and electromagnetic waveform increment range, while Figure 17b illustrates the current and electromagnetic field trend. It is noteworthy that the trend was not linear and that the box limited the current range. Lastly, Figure 18 illustrates the density and penetration distance of the electromagnetic waves in the various materials. It can be observed that lower densities allowed for greater transmission freedom. Figure 19a displays the working area of “B,” while (b) shows the penetration of “B” into the wall.

The term “B” refers to an electromagnetic wave with a value of 0.97x10^−6^ Tesla, which propagated through the sample materials, with a penetration depth greater than 30 cm, contingent on the presence of metals. The typical thickness of the walls was between 12 and 15 cm, regardless of the construction material used. When there was a strong relationship between density and porosity, it behaved as an attenuator to the induction of the electromagnetic waves, meaning that they penetrated less distance. It was observed that the electromagnetic field passed through the prepared walls without significant attenuation, and there was minimal current loss. 

The testing and examination of the prototype equipment on the walls revealed the presence of certain adjustment errors at the precise point where the equipment’s optimal working frequency (f) was situated. Consequently, the range of penetration of the magnetic field “B” was also optimal, independent of the specimen density. This examination is essential for the enhancement of the equipment.

## 4. Discussion 

The factor “B” is employed to gauge the depth of penetration, with the efficacy of this factor being enhanced by modulating the frequency and current, as previously outlined in reference [11]. Moreover, the designed and implemented magnetic and electrical property equipment detected structural failures depending on the depth of penetration and the density of the material, as previously mentioned in reference [47]. Additionally, in the experimental study [46], two specimens were tested, one comprising a poorly mixed concrete section and the other containing voids. However, the use of an electrical wire guide in conjunction with a reinforcement bar in the specimen led to the discovery that poor-quality concrete reduced the amplitude and velocity of the electromagnetic waves, a finding that aligns with the previously obtained results.

Further studies of a similar nature can be found in reference [48], in which a corrosion detection method for grounding networks is proposed. This method is based on the low-frequency electromagnetic method, which measures the resistance between individual nodes and facilitates the direct measurement of the induced magnetic field strength on the measuring conductor. It was observed that the magnetic field strength was stronger on a conductive network but not when measured on a non-conductive object. Conversely, in [49], an external tensor underwent a process of fine-tuning at the quantum critical point, resulting in notable alterations and the regulation of the optical characteristics. This finding partially corroborates the preceding assertion, as the input of (B) is capable of being calibrated according to the specific material and its thickness.

The law of the conservation of mass, as proposed by Antoine Laurent Lavoisier (1743–1794), states that the scalar magnitude cannot be created or destroyed, it can only be transformed. This is a relevant concept when comparing the density of building materials such as adobe, concrete and brick. In this regard, [50] posits that effective sustainability visions are characterised by brevity, clarity, future orientations, stability, challenges, abstraction and the capacity to inspire stakeholder satisfaction. Furthermore, the text recommends the inclusion of attributes such as unified love and good values.

Additionally, [51] posits that “polarization at interfaces represents a significant loss mechanism for the attenuation of electromagnetic waves (EMW)”. However, the attenuation of the electromagnetic field in our study pertained to the densities of the recently constructed materials, which underwent a loss of their original chemical and physical properties.

The prototype was employed in this study to ascertain whether, for concrete walls exhibiting a strength of ≥200 kg/cm^2^ or above, there would an increase in the frequency. In contrast, for concrete with a strength of 99 kg/cm² or below, a moderate decrease was observed. The findings also indicated that brick, both with and without voids, continued to demonstrate a decrease. Similarly, adobe and plaster exhibited a decrease. In the case of fine sand, the electromagnetic waves showed an insignificant decrease. 

It is essential to conduct comprehensive vulnerability assessments in low-maintenance and self-built dwellings, given that the vulnerability of these dwellings is influenced by soil characteristics, location and slope [34]. Additionally, the use of natural materials in construction is linked to a sustainable lifestyle, as discussed in [17]. However, it is important to note that common structural failures are often caused by inadequate concrete strength, insufficient bonding, excessive vibrations and torsional stresses. These observations are consistent with the issue addressed in this study [33]. The discussions are consistent with the evolution of the device, which addresses issues identified by multiple testers and has the advantage of not damaging walls, columns or ceilings when measuring their density. It provides a relationship between the penetrating power of the magnetic field and the consistency of the material, indicating that the magnetic potential is proportional to the viscosity of the material.

## 5. Conclusions

In the elaboration of the sustainable electromagnetic prototype developed in this study as a detector of the interior deterioration of walls present in buildings, algebraic calculations were employed with the listed equations, electronic circuit simulators and architectural plans, which provided results in anticipation of the hypothesis. Preliminary and final installations were carried out to analyse the operation of the prototype. Specimens were intentionally built with adobe material, concrete with resistances of 200 and 99 kg/cm^2^, respectively, brick with holes, plaster and fine sand. With the help of a metal reflecting guide on the opposite face, the following conclusions were reached: The electromagnetic field was demonstrated to have a strength of 0.18 µT, which allowed it to penetrate more than 300 mm. The density of the material in question fell between, 0.0–2.4 g/cm^3^, while the porosity ranged from 11–60%. The wall thickness of the material was 30 cm. A correlation was identified between the magnetic field strength, represented by the variable “B”, and the density of the material, represented by the variable “ρ”. This correlation is represented by the equation R = 0.9936, where R is the correlation coefficient. The relationship between these two variables was perfectly negative, indicating that as the density of the material decreased, the range of penetration of the electromagnetic wave increased.The generation of a suitable electromagnetic field, an inductance of 184 µH was required, a coil with several turns equal to 19. A positive correlation was observed between the inductance values calculated, measured, and produced by the software in micro-Henries (µH), where an increase in the number of turns of the wire resulted in an increase in the value of the inductance. Furthermore, the magnetic field in units of micro-Tesla (µT) also increased in line with the number of turns.An increase in the current (i) in transit collector Q1 resulted in a proportional increase in the induction of coil L1, which increased proportionally with a coefficient of determination R^2^ = 0.8914. This was exclusive to a 19-turn air- or vacuum-core coil with a radius of 9.75 cm and consisting of AWG number 25 enamelled copper wire.From the analysis of the amplitude modulation signal, it can be observed that the two signals exhibited a phase shift of 180° of the oscillation in the initial stage with respect to the second stage. The time required for one cycle of the waves was initially 85 µs, while the oscillation frequency was 10–22 kHz, which corresponded to an audio signal in the transducer.The refinement of the equipment was conducted in the laboratory using a range of electronic devices, including a digital oscilloscope, frequency meter, spectrum analyser, voltmeter and RCL meter. Precision readjustment was also employed, with the tests conducted on the specimens at 98%.Finally, the prototype was developed to be user-friendly, safe for the operator and environmentally sustainable. It is a cost-effective solution, offering rapid, appropriate and reliable performance while representing a significant engineering innovation compared to existing equipment.

## Figures and Tables

**Figure 1 sensors-24-04705-f001:**
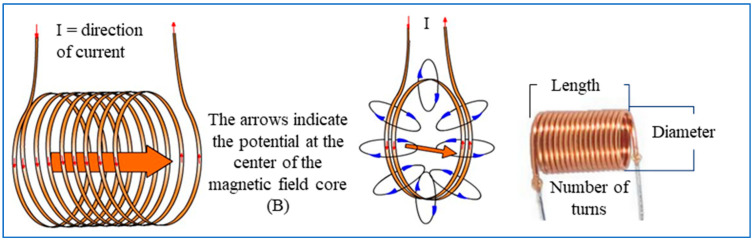
Coil and magnetic field (B). Note: The arrows indicate the direction of the current in the coil and the potential of the magnetic field at the center of the core.

**Figure 3 sensors-24-04705-f003:**
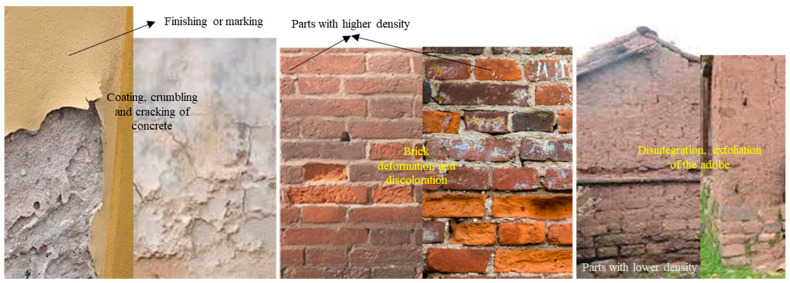
Structural deterioration in multiple constructions.

**Figure 4 sensors-24-04705-f004:**
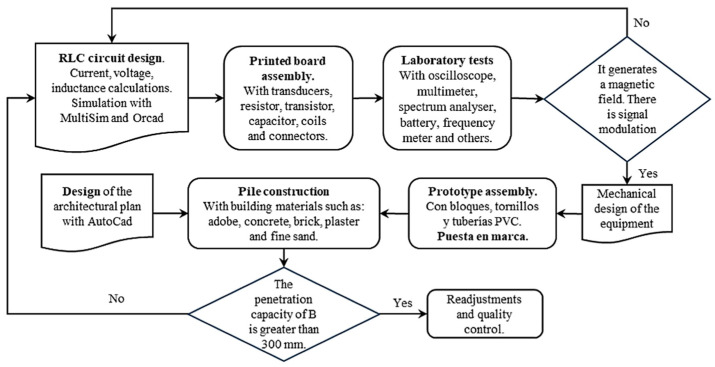
Prototype design and implementation process.

**Figure 5 sensors-24-04705-f005:**
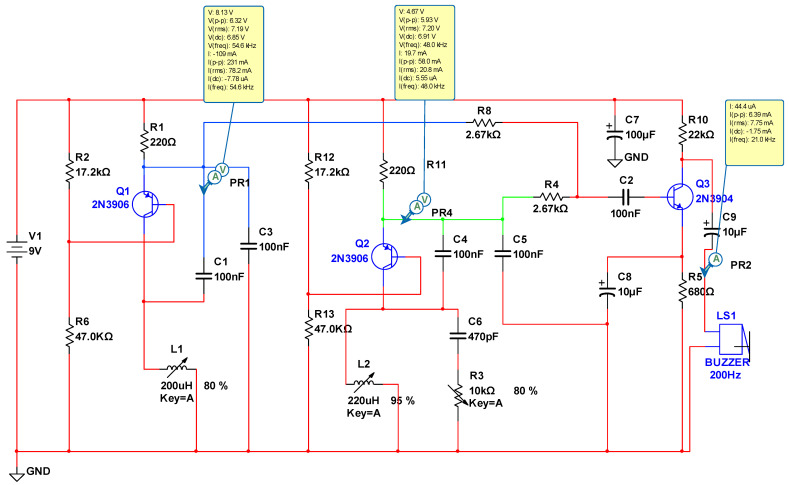
Basic design with MULTISIM of the electromagnetic field generator. Note: R = resistors, C = ceramic electrolytic capacitor, Q = transistors, L = inductance, LS = transducer, GND = ground, V = supply voltage.

**Figure 6 sensors-24-04705-f006:**
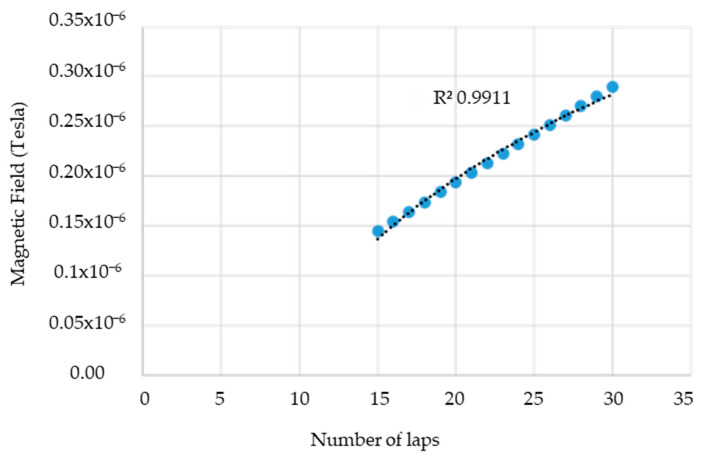
The ratio between the magnetic field and the number of spirals. Note: the coefficient of determination R^2^ = 0.9911, while Pearson’s linear correlation coefficient r = 0.9955, i.e., the calculations show that increasing the number of turns of the spiral increased the electromagnetic waves, but there are limits to everything.

**Figure 7 sensors-24-04705-f007:**
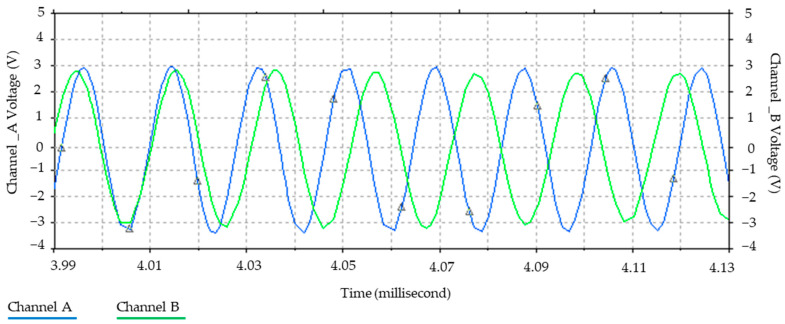
Oscillation frequency of transistors. Note: the channel A signal operated at 54.7 kHz and the channel B signal at 47.9 kHz. It can be seen that after 4.01 ms, both signals were in phase and then out of phase until they reached 180 degrees at 4.08 ms.

**Figure 8 sensors-24-04705-f008:**
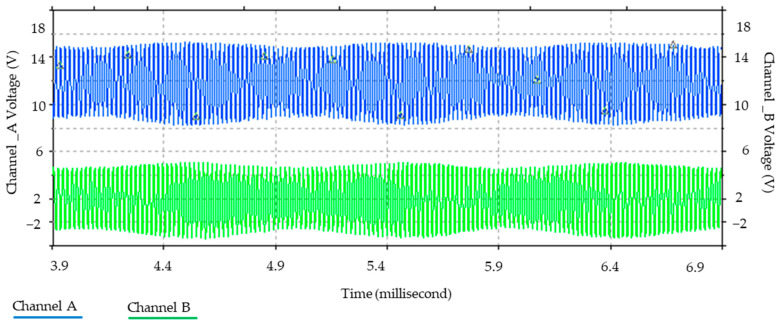
Amplitude and period of the oscillator waves. Note: amplitude (in volts) and period (in milliseconds) were monitored on channels A and B of the oscilloscope, so that transistors Q1 and Q2 in Figure 5 emitted oscillations at different frequencies.

**Figure 9 sensors-24-04705-f009:**
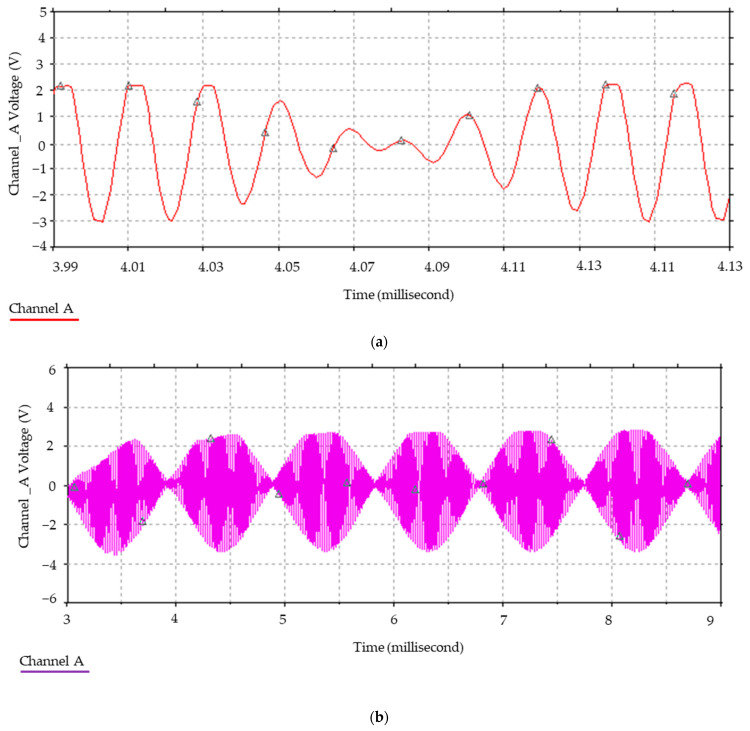
(**a**) Amplitude modulated waveform at the C2 input in Figure 5 and (**b**) carrier frequency from 2 to 22 kHz.

**Figure 10 sensors-24-04705-f010:**
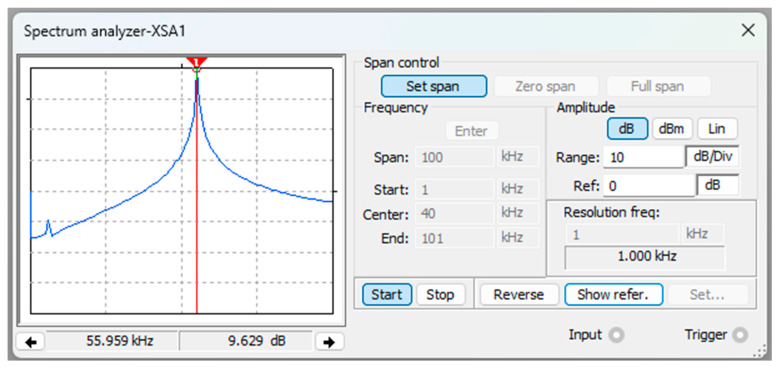
Electromagnetic spectrum analysis of the prototype. Note: the operating spectrum of the prototype was 56 kHz, with a gain of 10 dB.

**Figure 11 sensors-24-04705-f011:**
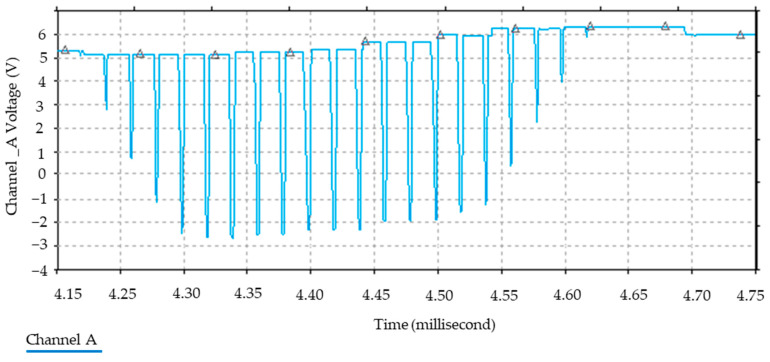
Audio carrier signal. Note: the light blue waves seen on the oscilloscope are the sound signal that leads to the final stage of the main circuit shown in Figure 5 and that is filtered before the transformer, where the switching depends on the change in inductance L1.

**Figure 12 sensors-24-04705-f012:**
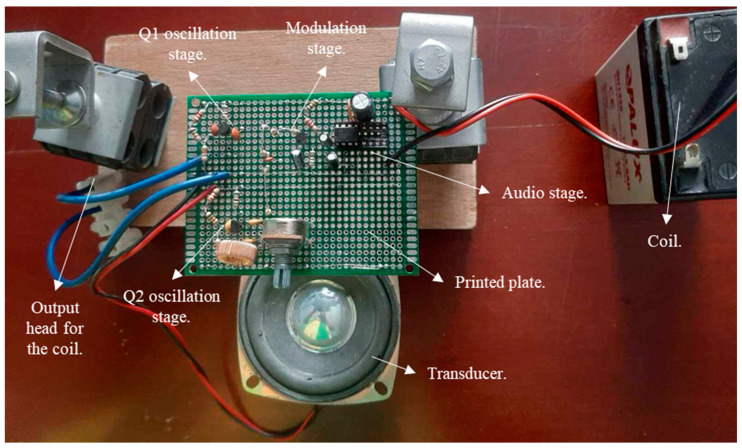
Internal parts of the electronic circuit of the prototype.

**Figure 13 sensors-24-04705-f013:**
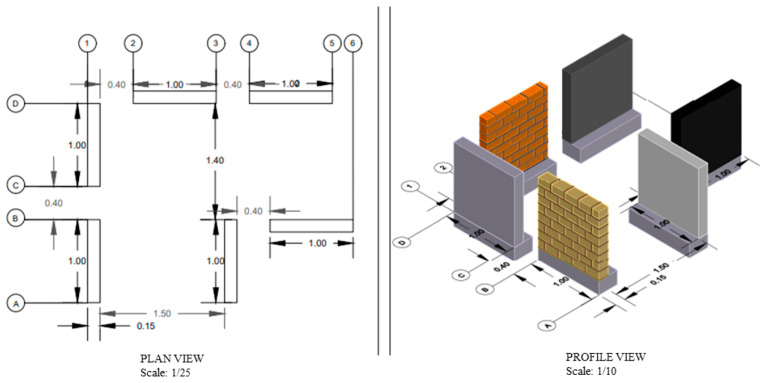
Specimen design. Note: a plan view can be seen on the left, while a perspective view can be seen on the right. The letters and numbers enclosed in circles represent the axes of the drawing.

**Figure 14 sensors-24-04705-f014:**
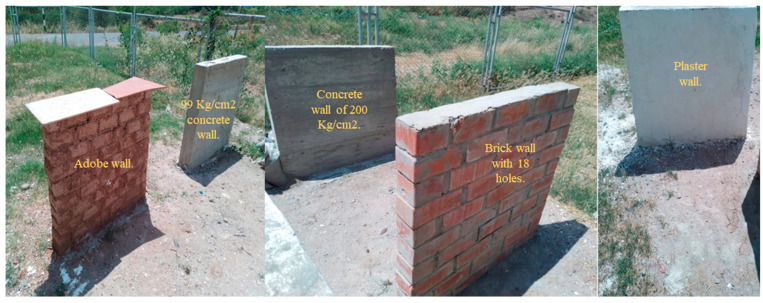
Specimens of different construction materials. Note: the specimens were constructed of adobe, brick, plaster, concrete and fine sand. Tests were conducted in open air “on specimens” one after the other, and this did not affect the results obtained because it was free from interference.

**Figure 15 sensors-24-04705-f015:**
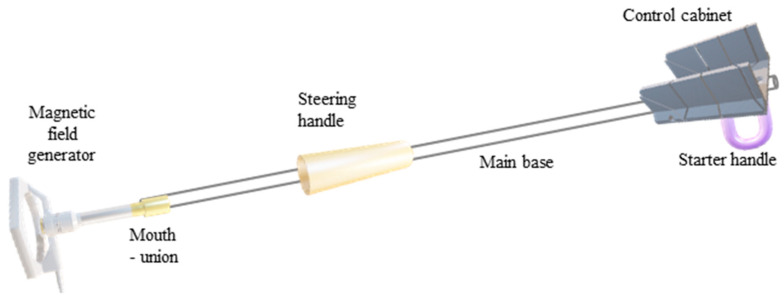
Prototype interior wall deterioration detector and its parts. Note: optimal performance and range tests were conducted on concrete, brick-with-holes, adobe, gypsum and fine-sand specimens.

**Figure 16 sensors-24-04705-f016:**
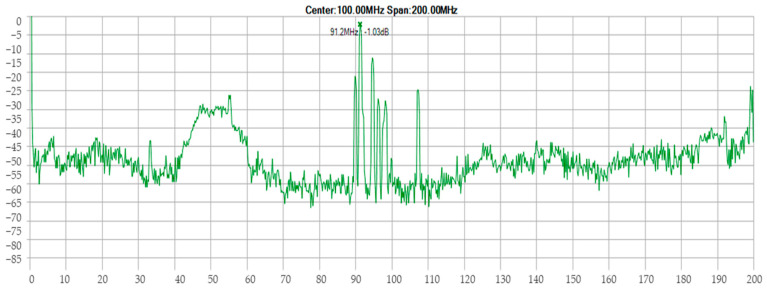
Analysis of the electromagnetic spectrum of the prototype. Note: the spectrum analyser shows the spectral components of the prototype; therefore, it is inferred that it was within the established non-ionising frequency range.

**Figure 17 sensors-24-04705-f017:**
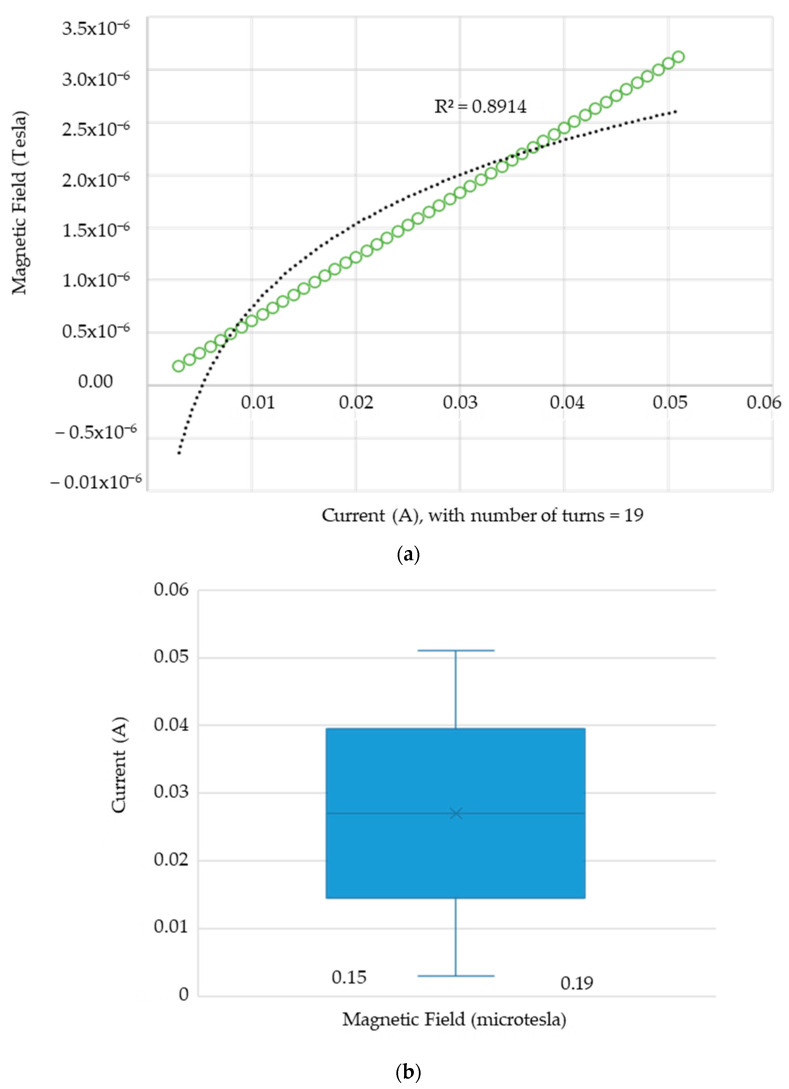
(**a**) Direction of current and electromagnetic field and (**b**) amplitude of current and electromagnetic waves. Note: as shown in (**a**), when the coil spiral number was 19, it was observed that the current direction in relation to the electromagnetic field was not linear, i.e., “>I⇒>B”, but it was limited at one point; no matter how much the current increased, the field increased very little. In addition, (**b**) shows a box-shaped shaded region, where its terminals indicate the normal current work between 15 and 38 mA and the magnetic field between 0.15 and 0.19 µt.

**Figure 18 sensors-24-04705-f018:**
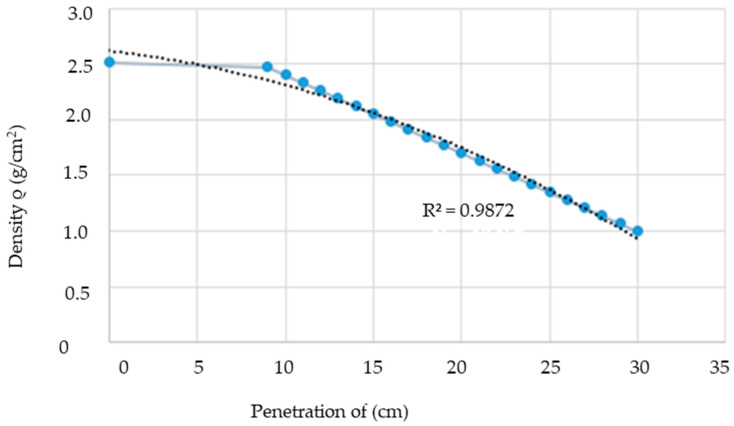
Density and penetration distance of electromagnetic waves. Note: the densities of adobe, concrete, brick, gypsum and fine sand were within the electromagnetic wave penetration range of the equipment; the lower the density, the greater the freedom of transmission, which reached the guide, the larger circles represent the relationship between density and field penetration distance, while the lines with smaller circles are the non-linear trend of these two parameters.

**Figure 19 sensors-24-04705-f019:**
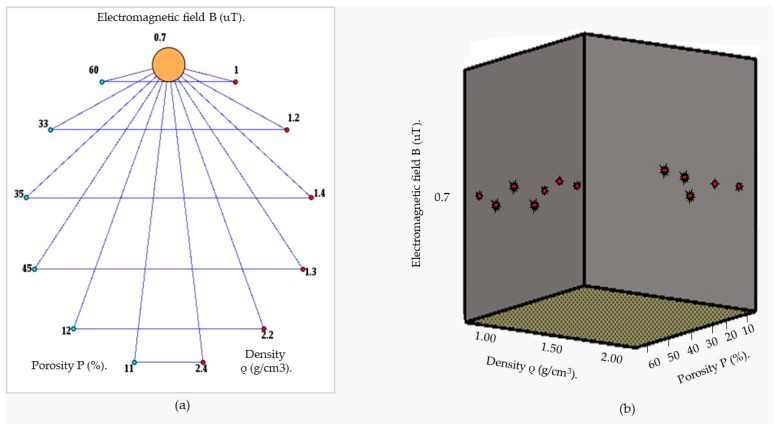
(**a**) Working area of “B”, (**b**) penetration of “B” into the wall.

**Table 1 sensors-24-04705-t001:** Methods used to assess infrastructure.

Masonry Structure	Methods	Advantages
With concrete	Ultrasonic pulse frequency	Adequate
With stone	Radar	Ideal
Brick and mortar	Radar or radication	Adequate
With chloride	Chloride test	Quick
Brick and mortar	Polarisation resistance	Recommended
With carbonation	Phenolphthalein indicator	Economical and simple
With concrete	Physical property identifier, UPV	Reliable
Faulty	Infrared thermography	Excellent
With brick	Destructive	Not recommended
Industrial sheet-metal segment	DIC 3D systems	Accuracy

Note: adapted from [3], added from [21] and supplemented from [24] for determining the position of cable breaks in long cable runs, where time domain reflectometers (TDR) are used.

**Table 2 sensors-24-04705-t002:** Density of construction materials.

Building Materials	Density ρ (g/cm^3^)
Wood	0.6–0.9
Water	1.0
Brick with holes	1.2
Cement	1.15–1.40
Adobe	1.2–1.7
Common baked brick	1.3–1.8
Compressed earth block	1.6–2.2
Fine sand	1.0–1.4
Concrete	2.2–2.5
Rock	2.7

Note: adapted from [46] for those used in the study test.

**Table 3 sensors-24-04705-t003:** Differences and inductance coefficients.

Number of Turns (Nv)	Lc (µH)	Lm (µH)	Ls (µH)
15	119.27	121.40	116.50
16	135.05	137.19	132.30
17	151.74	153.87	145.60
18	169.31	171.44	161.10
19	187.76	189.89	176.80
20	207.07	209.20	193.90
21	227.23	229.36	211.10
22	248.22	250.36	229.10
23	270.05	272.18	246.70
24	292.69	294.82	267.80
25	316.14	318.27	284.60
26	340.37	342.51	305.20
Correlation coefficient	r (Lc; Lm)	r (Lm; Ls)	r (Lc; Ls)
	1.0000	0.9998	0.9998

Note: calculated inductance is Lc, measured inductance is Lm, software inductance is Ls and the number of turns of the coil is Nv.

**Table 4 sensors-24-04705-t004:** Frequency results measured in each of the six specimens.

Specimen Materials	Frequency Range
Adobe	10–11
Concrete $f^{\prime }c$ = 99 kg/cm^2^	15–18
Concrete $f^{\prime }c$ = 200 kg/cm^2^	18–21
Perforated brick	14–15
Plaster	13–14
Fine sand	11–13

Note: frequencies that do not remain within the range may drop down or rise, depending on other factors such as humidity, temperature and metals, regardless of the material of construction.

## Data Availability

Data are contained within the article.

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
