# Peer review of "Sustainable Electromagnetic Prototype for Detecting Internal Deterioration in Building Walls"

_sensors, 2024, doi:10.3390/s24144705_

Round 1

Reviewer 1 Report

Comments and Suggestions for Authors

1. The abstract is not clear enough;

2. The font of ‘façade’ in the keywords is not standardized, and there are some words in the article with such problems;

3. Multiple references cited are not standardized, including Introduction, Materials and Methods, results and Discussion;

4. References and articles are not clearly expressed;

5. Multiple pictures and tables are not standardized, for example, Figure 9 description is not clear enough, Figures 4 and 15 are not clear enough, and the pictures and legends are in different pages, Tables 2 and 3 are in different pages;

6. Figure 12 each component should be clearly labeled;

7. Part of the paragraphs in the conclusion should not appear here, should be in RESULTS part;

8. Please pay attention to the citation format of references.

Comments on the Quality of English Language

Extensive editing of English language required

Author Response

En el archivo adjunto:

Reviewer 2 Report

Comments and Suggestions for Authors

This work on a magnetic sensor used to determine the structural strength of different materials using a lab setting is interesting.  But there are a few comments that should be addressed:

1) Equations should be numbered for reference, and the equation on line 98 is better referenced by "The Signal-to-Noise Ratio of the Nuclear Magnetic Resonance Experiment, Volume 24, Issue 1October 1976, Pages 71-85" as that is where reference 15 obtained the equation.

2) The equations used in section 1.1 should be used in context with the studied work or their role in similar works.

3) Some important ideas are very hard to understand: ""T" induced in the mass and angular momentum "u"" and equation line 145.

4)  What is FC and what are N and s representative of, other than coming from the Rockwell scale?

5) Paragraph at top of page 6 " [25] mentions that the State"...  has too many unrelated ideas that should be separated.

6) Figure 6: "Note: own elaboration" What does this mean?

7) The tests were performed outside: how comparable was the environmental conditions between different tests?

8) Why are the position of different walls from each other included in the CAD design?  Why weren't separate simulations performed to avoid crosstalk or intereference?

9) I don't see the results for "detection of interior defects" of these construction materials.

10) Discussion is not focused and not useful in explaining experimental results.

Comments on the Quality of English Language

1) Some expressions are unclear and could use a rewrite: "This design does not affect the aesthetics of the housing, i.e. free to destroy. The assembly process of the equipment is developed using materials, software / 3D program, and tests are performed on piles and 99% in the laboratory, allowing to properly assemble the prototype electromagnetic field (B) solves a latent problem that is the need of the family."

2) Some important ideas are very hard to understand: ""T" induced in the mass and angular momentum "u"" and equation line 145.

Author Response

En el archivo adjunto:

Reviewer 3 Report

Comments and Suggestions for Authors

In this study, the authors have developed a device that detects internal deterioration of coatings in adobe, brick, and concrete structures, preventing collapse, material, and economic loss. Experimental methods confirmed the validity of the method. Simulation and design accuracy provided 95% pre-test reliability. Five piles with different material constructions were involved in the prototype performance tests, and there was a correlation between the magnetic field range “B” (cm) and the density “ρ” (g/cm³) (R=0.9997), by increasing the overcurrent Q1 and the current (i) in coil L1, the induced intensity of “B” increases proportionally until the density penetrates the walls of the building with a reading error of 5%.

1.        In this manuscript, the abstract should clearly state the purpose and methodology of the study, as well as the innovation of the study. It is recommended that the authors revise the abstract section to be more concise and clearer so that the reader can quickly understand the general idea of the article.

2.        The authors should carefully check the formatting of the figure notes; the font of the figure notes in Figures 7 and 9 is different from the others. The author should be more rigorous in labeling the images.

3.        The authors should double check the formatting of the references inserted in the text, for example, there are errors in lines 342 and 343. This will make the article more rigorous.

4.        There are many grammatical errors in this manuscript, such as the end of line 68. The authors should have double checked the wording of the grammar of the article to be more scientific. add some references such as “J. Mater. Res. Technol., 2024, 29: 5667-5680., doi: 10.1002/adfm.202405523.”

5.        The authors should revise the conclusion section to be more concise; the conclusion should briefly answer the research question and point out the significance of the study, and finally summarize the shortcomings of the study.

Comments on the Quality of English Language

minor 

Author Response

En el archivo adjunto:

Round 2

Reviewer 1 Report

Comments and Suggestions for Authors

accept

Reviewer 2 Report

Comments and Suggestions for Authors

Changes made are sufficient and acceptable.